# Comparison between self-reported and pedometer-measured physical activity in Vietnamese adolescents: A reliability and agreement study

Tram T. N. Truong(iD), Van-Anh N. Huynh(iD), Kien G. To(iD)*

Faculty of Public Health, University of Medicine and Pharmacy at Ho Chi Minh City, Ho Chi Minh City, Vietnam

* togiakien@ump.edu.vn

## Abstract

To compare the level of physical activity (PA) measured by self-reported questionnaire and pedometer in 12-year-old Vietnamese adolescents. Data was collected from March to May 2023 in two phases among 510 sixth-grade students in Ho Chi Minh City, Vietnam. The first phase involved a pilot study, which recruited 35 students from a randomly selected class to evaluate the Physical Activity Questionnaire for Older Children (PAQ-C) test-retest reliability. PA was measured twice, with seven-day intervals between assessments. The second phase was a cross-sectional study including 475 sixth-grade students to evaluate the agreement between PAQ-C scores and pedometer data. Participants wore a pedometer continuously for seven days and completed the PAQ-C at the end of the monitoring period. Sociodemographic and anthropometric data were collected. The sample of 35 students showed that PAQ-C demonstrated good internal consistency ($\alpha = 0.89$) and test-retest reliability (ICC = 0.82). After one week, McNemar's test indicated no significant difference in activity levels between the baseline and second assessment. In the sample of 475 students, there was a substantial agreement between PAQ-C and pedometer in classifying physical activity levels (Cohen's $\kappa = 0.688$, $p < 0.001$), with a strong correlation between daily step counts and PAQ-C scores ($r_s = 0.855$, $p < 0.001$). PAQ-C is a reliable, cost-effective tool for school-based PA monitoring in 12-year-old Vietnamese adolescents, particularly in resource-limited settings.

## Introduction

Physical activity (PA) is crucial for reducing the incidence of non-communicable diseases and promoting good health and well-being, especially during adolescence [1,2]. However, the prevalence of insufficient PA among adolescents was alarmingly high, with approximately 80% not engaging in enough PA [3]. In 2016, World Health

**Data availability statement:** All relevant data are within the paper and its Supporting Information files.

**Funding:** The author(s) received no specific funding for this work.

**Competing interests:** The authors have declared that no competing interests exist.

Organization (WHO) reported that approximately 78% of boys and 84% of girls between the ages of 11 and 17 did not meet the recommended levels of PA [3]. Activities among adolescents in developing countries, like Vietnam, were often overlooked [4]. In Ho Chi Minh City, most students lead sedentary lifestyles due to their dependency on motorized transportation, extensive studying, excessive use of digital technology, and consumption of high-calorie food [5,6]. There is a pressing need for more research and initiatives to promote PA among adolescents in developing countries.

Accurately measuring PA is a complex and challenging task in adolescents. While many objective PA assessment techniques exist, such as pedometers, can provide accurate and useful data for measuring PA in adolescents, particularly step counts. A combination of pedometers and self-reported questionnaires in measuring PA is more efficient and provides more accurate data [7,8]. However, in the resource-limited and time-constrained contexts, self-reported questionnaires are more suitable to measure PA due to their cost-effectiveness, ease of implementation, and ability to quickly gather a wide range of information [9].

Self-reported questionnaires, such as the Physical Activity Questionnaire for Older Children (PAQ-C), the Global Physical Activity Questionnaire (GPAQ), and the International Physical Activity Questionnaire-Short Form (IPAQ-SF), were the most commonly used subjective methods for collecting data on PA [10,11]. On the other hand, the PAQ-C is a simple assessment of general PA levels, which is one of its strengths, unlike the GPAQ or IPAQ questionnaire, it is difficult to precisely measure the intensity, frequency, and duration of young people's activities [12]. Biddle et al. reviewed available self-report PA instruments developed for children and adolescents to assess their suitability and feasibility. As part of this assessment, five international experts in PA measurement were invited to critically appraise a shortlist of 20 activity-based measures. The evaluation process involved a detailed examination of key psychometric properties, such as reliability, validity, and ease of administration. Based on expert consensus, the PAQ-C was identified as one of the three measures that received substantial support, demonstrating robust measurement properties and practical applicability for assessing PA in youth populations [13]. Therefore, we chose the PAQ-C as the PA self-report assessment method for children in this study.

Several studies explored the relationship between pedometer data and subjective self-reported measures of PA. For instance, McGee and colleagues discovered a moderate correlation between daily step count and the self-reported PA questionnaire in 4th and 5th grade students [14]. Similarly, Wilde and colleagues found a correlation between pedometer step count and reported levels of PA [15]. However, self-reported assessments were limited by information bias and respondents' misunderstandings of the questions [16]. Although accelerometers can collect more comprehensive data such as intensity, duration and type of PA, their high cost and technical complexity make them less accessible in resource-limited countries like Vietnam. Pedometers are cheaper and easier to use, making them more practical for larger studies and helping participants use them more effectively [16]. While pedometers do not capture PA intensity or duration, they accurately measure step counts and have demonstrated strong correlations with uniaxial accelerometers and moderate-to-vigorous PA

[17,18]. Moreover, the steps/day metric is considered a valid indicator of habitual PA [19]. Despite their limitations, pedometers are recognized as reliable and accurate tools for assessing PA in children and adolescents [7] Among the available methods, there was limited evidence for determining whether someone was physically active or not, especially in Vietnamese adolescents. This study evaluated the validity and reliability of the Vietnamese version of the PAQ-C by comparing it with pedometer-based measurements in 12-year-old students. By examining the agreement between these methods, we aimed to demonstrate the feasibility of using the PAQ-C as a practical, low-cost alternative for widespread PA assessment and school-based interventions.

## Materials and methods

### Ethics statement

The study was approved by the Ethics Committee, University of Medicine and Pharmacy at Ho Chi Minh City (261/HDDD-DHYD, 3rd March 2023). Information sheets were sent to all students and their parents or legal guardians. Written informed consent was obtained from parents or legal guardians, and assent forms were collected from students before data was collected.

### Study's design and setting

The study was conducted in two phases. The first phase consisted of a pilot study to assess the test-retest reliability of the self-reported questionnaire. The second phase involved a cross-sectional study to evaluate the agreement between the questionnaire scores and objective pedometer data. Data collection for both phases was performed from March to May, 2023, at Nguyen Du Secondary School, Go Vap District, Ho Chi Minh City, Vietnam. Ho Chi Minh City, the economic center of southern Vietnam, is a densely populated metropolis. Compared to other regions in Vietnam, the city has a higher prevalence of childhood overweight and obesity, likely influenced by urban lifestyles, dietary patterns, and reduced PA levels [6]. This data collection period was chosen as it represents a stable climate year-round in Ho Chi Minh City, with minimal extreme weather events that could significantly impact outdoor activities. Moreover, this timeframe aligned with the second semester, when students followed a structured academic schedule with regular physical education classes and recess. The majority of data was gathered before final examinations to mitigate potential declines in PA due to increased academic workload.

Go Vap District is one of the most rapidly urbanizing districts in the city and the second most populous, with 602,180 inhabitants [20]. Nguyen Du Secondary School, a public institution in Go Vap District, was selected for this study due to its large class sizes, and students were willing to cooperate with their physical education teachers who received pedometer training. The school enrolled 1,951 students across four grade levels, with class sizes ranging from 45 to 49 students per class. The school follows the curriculum set by the Vietnamese Ministry of Education and Training. The curriculum includes two physical education classes per week and a sports activity session every Saturday morning. The school also offers extracurricular sports clubs, like soccer and basketball, and organizes sports competitions twice a year.

This study adhered to the Guidelines for Reporting Reliability and Agreement Studies (GRRAS) to ensure comprehensive reporting of the measurement properties analysis [21] (see S1 Checklist).

### Study's participants

The sixth grade students, who aged 12 and over, were targeted in this study because they were in the early stages of adolescents. At the age of 12, children typically go through a stage of rapid physical development, specifically in height and weight, along with changes in body structure and sexual characteristics. Interventions during this time could have a positive impact on bone density, muscle development, and overall physical fitness [22]. At this age, children have cognitive developments that define their individual consciousness and ideals, including attention and decision-making [22,23].

Therefore, interventions for 12-year-old children provide positive impacts of PA on their cognitive development healthy behaviours. Children at this age are influenced by their peers; therefore, group activities and team sports could be particularly beneficial, promoting social interaction, teamwork skills, and effective communication [22,24,25].

In Vietnam, Secondary School comprised four grades from sixth to ninth for students aged 12–15. The study focused on sixth grade students as they were the youngest in Secondary School. Providing information about PA status of this group would help schools develop appropriate interventions for the following years.

## Sample size and sampling

The study recruited all 552 sixth-grade students across 12 classes at Nguyen Du Secondary School. In the first phase, a pilot study was conducted with a randomly selected sample of 35 students from one class to test the measurement tools. This sample size was based on previous validity and reliability studies, which recommend that 18–50 participants are typically sufficient for estimating Intraclass Correlation Coefficients (ICCs) to assess the validity between two methods [26]. These students were assessed twice, seven days apart, which is a period commonly used to evaluate behavioural consistency [27]. For the second phase, sample size calculations followed the formula for prevalence studies by Naing et al. [28], with a design effect (deff) of 1.5 to adjust for clustering [29]. The prevalence of low physically active was 0.24 [30]. Using a 95% confidence level (Z = 1.96) and a precision (d) of 0.05. The calculated sample size was 422. All 517 sixth-grade students from 11 remaining classes were invited to participate in the cross-sectional survey, with each class serving as a cluster unit. A self-reported questionnaire and pedometer were distributed and collected by the research team during physical education classes. Students with disabilities or acute or chronic illnesses that would affect their participation in the study and biometric measurements were excluded from the study.

## Data collection and tools

Gender and age were self-reported by students. Height and weight were measured using the WHO's methods [31]. Height was measured using a stadiometer (model 26SM, Tamil Nadu, India) with an accuracy of 0.1 cm, while weight was measured using an electronic scale (model HD 379, Tanita Corporation, Japan) with an accuracy of 0.1 kg. Each measurement was taken twice by a trained researcher, with the students in an upright position, barefoot, and wearing light clothing. The averages of the two measurements were used to ensure accuracy. Body Mass Index (BMI) was calculated as weight (kg) divided by height (m) squared. Students were classified based on the z-score of BMI using the WHO Child Growth Standards [32].

**The Physical Activity Questionnaire for Older Children (PAQ-C).** PAQ-C is a self-administered instrument that utilizes a 7-day recall period to assess general levels of PA in children [33]. PAQ-C has shown good reliability with young individuals aged 8–14 [9,34,35]. This questionnaire consists of 10 items, with the first item being a checklist of the frequency of engaging in activities during spare time. The next six items assess PA during specific time periods: physical education classes, recess, lunchtime, after school, evening, and weekends. The eighth item evaluates the frequency of PA in the past 7 days, and the ninth item is a checklist of the frequency of PA on each day of the week. Although the tenth item identifies PA's barriers, it is not included in the overall score. Each item is scored from 1 (low PA) to 5 (high PA), and the total PAQ-C score is calculated by averaging all nine item scores. A total PAQ-C score of ≤2.75 indicates insufficient activity, whereas a score of >2.75 represents sufficient activity [36]. Participants who completed <75% of the questionnaire were excluded from the analysis.

PAQ-C was translated into Vietnamese by a member of the research team who was a native Vietnamese speaker with fluency in English. Subsequently, a Vietnamese translator residing in the United States translated the Vietnamese version back into English. Both versions were reviewed by the research team and a native English speaker to ensure linguistic and cultural equivalence. After extensive discussion, certain activities listed in the first item were removed since they were less common in Vietnam (e.g., rowing/canoeing, in-line skating, tag, swimming, baseball, softball, dance, football,

skateboarding, street hockey, floor hockey, ice skating, cross-country skiing, ice hockey/ringette), and replaced by more common activities such as jumping rope, martial arts, and shuttlecock kicking. As a result, the activity checklist was condensed to 12 activities instead of 22 in the original version. These activities have been similarly used and are widely accepted to assess the proportion of PA among Vietnamese children in previous studies [6,37,38].

PAQ-C was administered to assess students' PA after seven days of step count monitoring. This approach allowed for the evaluation of students' perceived PA during the precise timeframe when the pedometer recorded step counts, facilitating direct comparisons between subjective self-reports and objective data. Previous studies have employed similar methods to align the recall period with the monitoring timeframe [14,39,40]. After the final day of step count, students completed the PAQ-C at their classroom.

**Pedometer.** An unsealed commercial pedometer with an accuracy of approximately ±10% was used to assess PA of students. This small device senses body motion, counts steps in real time, and has demonstrated acceptable validity with the PAQ-C in adolescents [39].

On the first day, students were instructed on how to use the pedometer, which was attached to their belts from the moment they woke up until they went to sleep, except during activities like swimming or bathing. At the end of each day, students recorded their daily step count on a diary sheet and reset the pedometer the next morning. Although some concerns exist regarding measurement reactivity when students see their daily step totals, previous studies have indicated that this does not significant threaten measurement accuracy [41,42]. However, to minimize potential bias, students were initially allowed to shake the pedometer to observe its response but were then instructed not to repeat this behavior in the following days. This approach has been shown to reduce excitement and curiosity among children [43]. In addition, students were required not to focus on the step count display or modify their PA based on the numbers shown. Monitoring lasted for seven days, during which researchers visited each class every morning to remind students to wear the pedometer, collect diary sheets, and check recorded step counts for inconsistencies. Any abnormal step count data were carefully examined and either verified at that time. Values less than 1,000 steps/day or greater than 30,000 steps/day were considered invalid and excluded, following established criteria from previous studies [43–45]. These cut-off points were chosen based on their alignment with hypothetical scenarios of the "least active child" and "most active child", and represented 1% of the data points, ensuring a practical and reasonable objective criterion. Additionally, rounding the lower limit to the nearest 1,000 steps provides values that are easier to remember for data analysis purposes [43].

Students who forgot to track their steps on any weekday (from Monday to Friday) were instructed not to wear the pedometer for the entire day. In such cases, a value of 0 was recorded on the diary sheet and then replaced with the average step count for the other weekdays [46]. For cases with missing data on the weekend (Saturday or Sunday), the step count was replaced with data from the remaining weekend day. However, if step counts were missing on both Saturday and Sunday, they were excluded from the analysis [46]. PA levels were classified according to Rowe et al., with <10,000 steps/day indicating insufficient activity and ≥10,000 steps/day considered sufficient activity [43]. The ICC for step counts was calculated for a single day, followed by determining the number of days needed to achieve a reliability of 0.7 or higher using the Spearman-Brown prophecy formula [47–49]. Pedometer-determined PA was considered valid if the participant wore the device for at least three days, including any two weekdays and one weekend day during the seven-day monitoring period.

## Statistical analyses

Data were analyzed using Stata version 14.0 (StataCorp, College Station, TX, USA) and computed separately for the two samples (see S1 Data). Descriptive statistics (i.e., frequencies, percentages, means, standard deviations, medians, and interquartile ranges) were computed for participant characteristics. Independent t-test or Mann-Whitney U test was done to examine differences in continuous variables between the two samples. Differences in categorical variables were assessed using Pearson Chi-squared test.

The internal consistency of PAQ-C was evaluated using Cronbach's Alpha (α). Values of α greater than 0.7 were considered acceptable for research purposes [50]. Test-retest reliability of each question was assessed using the ICC for the first sample of 35 students. The two-way mixed effects model and the absolute agreement type were used to calculate the ICC between the two assessments.

The two-by-two cross-tabulation table was created to compare PAQ-C and pedometer in classifying PA levels. Cohen's Kappa test was calculated to measure the level of agreement between the two methods. The strength of agreement was interpreted as follows: ≤ 0.20 as none to slight, 0.21–0.40 as fair, 0.41–0.60 as moderate, 0.61–0.80 as substantial, and 0.81–1.00 as almost perfect agreement [51]. McNemar's test was performed to check for any differences in the PA levels determined by PAQ-C and pedometer between the baseline and the second assessment after one week. Spearman's rank correlation coefficient ($r_s$) was used to analyze the validity of the daily step count and the PAQ-C scores. All p-values <0.05 were considered significant.

## Results

The researcher approached 552 students. Thirty-five students were invited to participate in the pilot study for test-retest reliability, while 517 students participated in the cross-sectional survey. Amongst the 517 students, 18 refused to participate, 15 refused to wear pedometers, and nine students did not complete the questionnaires, resulting in the final sample of 475 students. The entire process of the study analysis is presented in Fig 1.

Table 1 displays the characteristics of the study participants across the two phases. The mean age was 11.8±0.2 years in the first sample (N = 35) and 11.8±0.3 years in the second sample (N = 475). The gender distribution was comparable between the two samples, with 51.4% boys and 48.6% girls in the first sample and 50.7% boys and 49.3% girls in the second sample. Overall, there were no significant differences in demographic characteristics and PA levels between the two samples, except for height (p = 0.013) and BMI (p = 0.008). Less than 50% of students met the criteria for sufficient activity when assessing PA using both PAQ-C and pedometer methods.

After one week, the results from Table 2 for the sample of 35 students show no significant difference in the percentage of respondents with sufficient versus insufficient activity levels between the baseline and the second assessment.

Table 3 shows that PAQ-C had acceptable internal consistency for the first sample (n = 35) in the two assessments and the second sample (N = 475), with Cronbach's Alpha coefficients of 0.89, 0.86 and 0.89, respectively. Moreover, PAQ-C had high reliability with an Intra-Class Correlation Coefficient (ICC) of 0.82 (95% CI = 0.48-0.93). The inter-item reliability within the items ranged from moderate to very high, with the lowest ICC being 0.57 (95% CI = 0.17-0.79) for the activity checklist, while the highest ICC was 0.96 (95% CI = 0.92-0.98) for PA right after-school.

The total PAQ-C score exhibited a high correlation with the daily step count measured by a pedometer among 475 adolescents ($r_s$ = 0.855, p<0.001), as presented in Fig 2.

Among the 475 students, there were more physically active respondents in the boys' group and the normal weight group for both assessment methods. Approximately 40.7% of boys met the criteria for sufficient activity based on PAQ-C, while 34.0% met the criteria using the pedometer method (Table 4). Cohen's Kappa was used to determine the agreement between two methods in assessing the level of PA among respondents. The results showed a substantial agreement between PAQ-C and pedometer methods (Cohen's κ = 0.688, p<0.001), as presented in Table 5.

## Discussion

This study was conducted in a high school in Ho Chi Minh City, Vietnam, that assessed the reliability and validity of pedometer data and a self-reported PA questionnaire. The PAQ-C had good reliability and a strong correlation with the daily step count measured by the pedometer. In fact, our coefficients were higher than those reported by other validation studies in adolescents [40,52,53].

PAQ-C was a popular and efficient tool for assessing PA in adolescents [13]. Our results showed that PAQ-C had good internal consistency (α = 0.89) that is consistent with previous studies. For instance, Croker et al. reported good internal

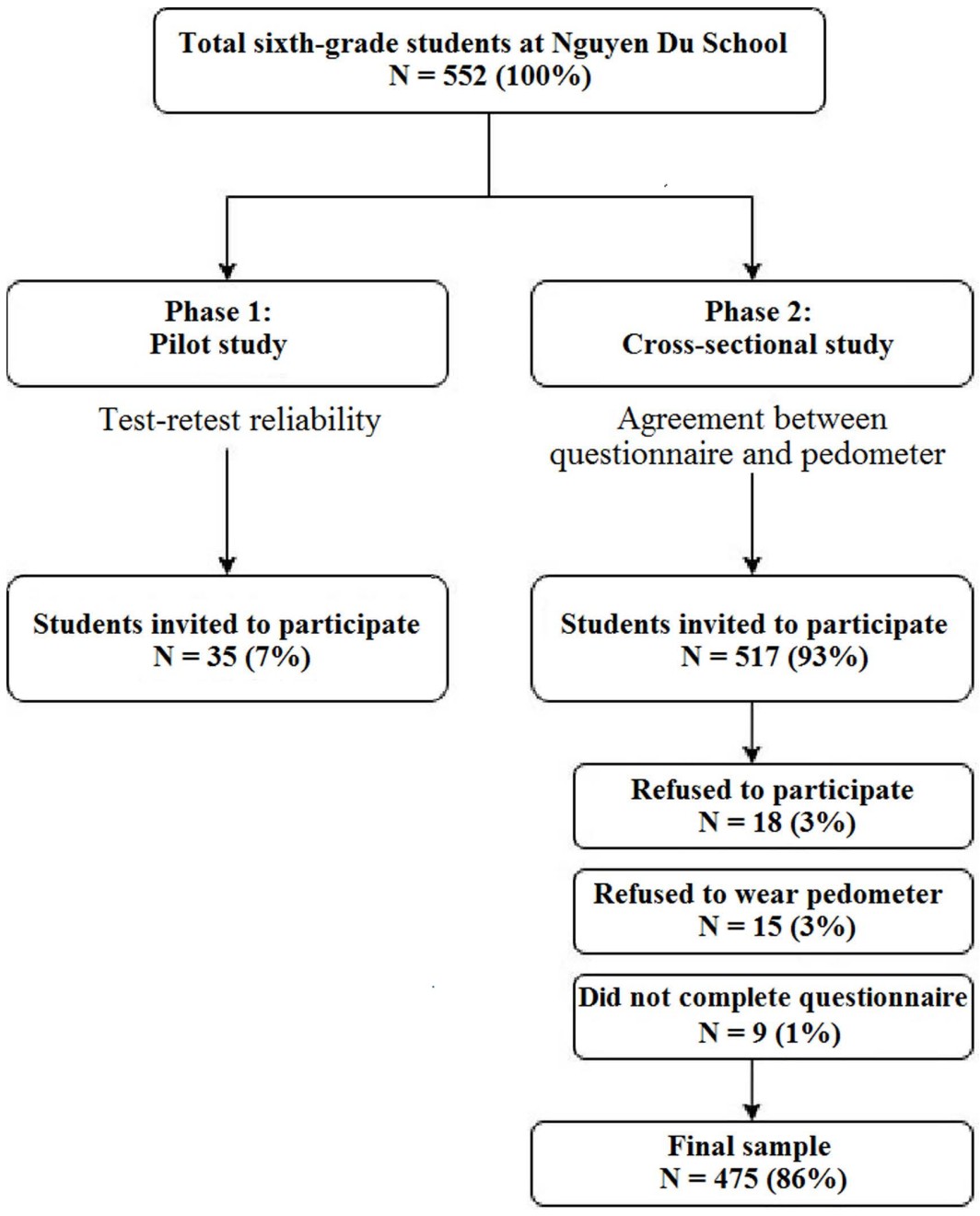

**Fig 1. The flow diagram of the study.**

consistency of PAQ-C with α = 0.79 and α = 0.89 for the first and second administrations, which was one week apart from one another [54]. Similarly, Kathleen and colleagues found good internal consistency of α = 0.72 in a study with 210 11-year-old children, suggesting its utility for both the school year and summer vacation [55]. Another study conducted with a sample of 36 13-year-old students also demonstrated high internal consistency of PAQ-C [56]. For test-retest reliability, the study showed the stability of PAQ-C over time with ICC = 0.83 (95% CI: 0.49-0.93) for the total score. These findings were higher than a study conducted in Vietnam among 16-year-old students [57], but similar to studies in Turkey

Global Public Health

**Table 1. Demographic characteristics of study's participants.**

| Characteristics | Description | First sample (N = 35) | Second sample (N = 475) | p-value |
|---|---|---|---|---|
| **Age** (years) | Mean±SD | 11.8±0.2 | 11.8±0.3 | >0.999[a] |
| **Height** (centimetres) | Mean±SD | 149.9±6.0 | 152.6±6.2 | **0.013[a]** |
| **Weight** (kilograms) | Mean±SD | 45.8±6.6 | 44.5±7.4 | 0.313[a] |
| **BMI** (kg/m²) | Mean±SD | 20.4±2.7 | 19.1±2.8 | **0.008[a]** |
| **Gender** | n (%) | | | 0.937[b] |
| Boy | | 18 (51.4) | 241 (50.7) | |
| Girl | | 17 (48.6) | 234 (49.3) | |
| **Weight status** | n (%) | | | 0.071[b] |
| Underweight/ Normal weight | | 19 (54.3) | 328 (69.0) | |
| Overweight/ Obesity | | 16 (45.7) | 147 (24.0) | |
| **Total PAQ-C score** | Median (IQR) | 2.6 (2.0 – 3.3) | 2.4 (2.0 – 3.0) | 0.475[c] |
| **Pedometer** (steps/day) | Median (IQR) | 7,919.3 (5,450.3 – 13,166.9) | 7,437.1 (5,205.6 – 10,908.4) | 0.164[c] |
| **PA level by PAQ-C** | n (%) | | | 0.295[b] |
| Sufficient | | 16 (45.7) | 175 (36.8) | |
| Insufficient | | 19 (54.3) | 300 (63.2) | |
| **PA level by pedometer** | n (%) | | | 0.530[b] |
| Sufficient | | 13 (37.1) | 152 (32.0) | |
| Insufficient | | 22 (62.9) | 323 (68.0) | |

BMI: Body Mass Index; PAQ-C: Physical Activity Questionnaire for Older Children; PA: physical activity; SD: Standard Deviation; IQR: Inter-Quartile Range; n: frequency; %: percentage.

[a]Independent t-test;

[b]Pearson Chi-squared test;

[c]Mann-Whitney U test.

**Table 2. Changes in physical activity level measured by PAQ-C and pedometer from baseline to second assessment (N = 35).**

| Test | Baseline assessment | Second assessment | p-value |
|---|---|---|---|
| | n (%) | n (%) | |
| **PAQ-C** | | | 0.317[d] |
| Sufficient activity | 16 (45.7) | 14 (40.0) | |
| Insufficient activity | 19 (54.3) | 21 (60.0) | |
| **Pedometer** | | | 0.157[d] |
| Sufficient activity | 13 (37.1) | 9 (25.7) | |
| Insufficient activity | 22 (62.9) | 26 (74.3) | |

PAQ-C: Physical Activity Questionnaire for Older Children.

[d]p-value was calculated using McNemar's test.

[39], Malaysia [58], and China [35]. Most studies have found that PAQ-C is a reliable and stable tool over time, and differences in sample size and age of participants may affect the results [39].

The validity of pedometer and PAQ-C data were determined by correlating the two methods. While pedometers recorded the number of steps taken daily, they did not capture the intensity, frequency and type of PA. However, they were suggested as one of the best measures for assessing PA in children [59]. We found a strong correlation between PAQ-C and the step count recorded by pedometer in children ($r_s = 0.855$; $p < 0.001$). After one week, the level of sufficient activity measured by both methods decreased. This might be because students were having their final exams during this

**Table 3. Descriptive statistics, internal reliability and test-retest reliability for PAQ-C.**

| Variables | First sample (N=35) | | | | | Second sample (N=475) | |
|---|---|---|---|---|---|---|---|
| | Baseline assessment | | Second assessment | | ICC (95% CI) | Formal assessment | |
| | Median (IQR) | Mean±SD | Median (IQR) | Mean±SD | | Median (IQR) | Mean±SD |
| **Total PAQ-C score** | 2.6 (2.0 − 3.3) | 2.6±0.8 | 2.5 (1.9 − 3.1) | 2.5±0.7 | 0.82 (0.48; 0.93) | 2.4 (2.0 − 3.0) | 2.5±0.7 |
| Q1. Activity checklist for spare time | 2.0 (1.4 − 2.4) | 2.1±0.8 | 1.7 (1.3 − 2.0) | 1.7±0.4 | 0.57 (0.17; 0.79) | 2.0 (1.6 − 2.4) | 2.1±0.6 |
| Q2. Activity in physical education classes | 4.0 (3.0 − 4.0) | 3.8±0.8 | 4.0 (3.0 − 4.0) | 3.7±0.9 | 0.77 (0.59; 0.88) | 4.0 (3.0 − 4.0) | 3.7±0.9 |
| Q3. Recess activity | 2.0 (1.0 − 3.0) | 2.1±1.2 | 2.0 (1.0 − 2.0) | 2.1±1.1 | 0.92 (0.85; 0.96) | 2.0 (1.0 − 3.0) | 2.2±1.1 |
| Q4. Lunchtime activity | 2.0 (1.0 − 2.0) | 2.0±1.2 | 2.0 (1.0 − 3.0) | 1.9±1.0 | 0.77 (0.58; 0.88) | 2.0 (1.0 − 3.0) | 2.1±1.0 |
| Q5. After-school activity | 2.0 (2.0 − 3.0) | 2.5±1.0 | 2.0 (2.0 − 3.0) | 2.6±1.0 | 0.96 (0.92; 0.98) | 2.0 (1.0 − 3.0) | 2.3±1.2 |
| Q6. Evening activity | 2.0 (1.0 − 4.0) | 2.4±1.5 | 2.0 (1.0 − 3.0) | 2.3±1.2 | 0.93 (0.86; 0.96) | 2.0 (1.0 − 3.0) | 2.2±1.1 |
| Q7. Weekend activity | 3.0 (2.0 − 4.0) | 2.8±1.3 | 3.0 (2.0 − 4.0) | 2.8±1.1 | 0.87 (0.76; 0.93) | 2.0 (2.0 − 3.0) | 2.5±1.2 |
| Q8. Describe activity frequency in the last 7 days | 3.0 (1.0 − 4.0) | 2.7±1.4 | 3.0 (2.0 − 4.0) | 2.7±1.2 | 0.93 (0.87; 0.97) | 3.0 (2.0 − 4.0) | 2.7±1.2 |
| Q9. Frequency activity on each day last week | 3.0 (2.4 − 3.4) | 2.9±0.9 | 2.7 (2.3 − 3.4) | 2.8±0.8 | 0.90 (0.79; 0.95) | 2.7 (2.0 − 3.6) | 2.8±1.0 |
| **Cronbach's Alpha** | 0.89 | | 0.86 | | | 0.89 | |

IQR: interquartile range; SD: standard deviation; ICC: intra-class correlation coefficient; 95% CI: 95% confidence interval; PAQ-C: Physical Activity Questionnaire for Older Children.

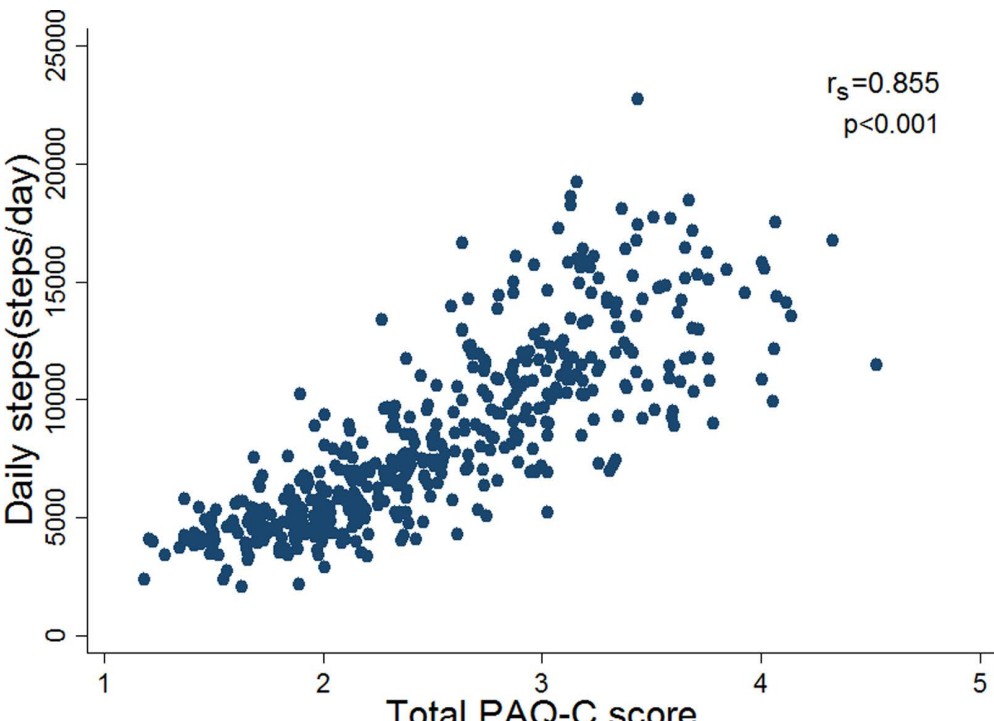

**Fig 2. Scatter plot of daily step count against the total scores in the PAQ-C questionnaire (N=475).** The Fig 2 legend: $r_s$ = Spearman's rank correlation; PAQ-C = Physical Activity Questionnaire for Older Children.

**Table 4. Physical activity level by PAQ-C and pedometer categorized by gender and weight status (N = 475).**

| Variables | PAQ-C | | Pedometer | |
| --- | --- | --- | --- | --- |
| | Sufficient activity | Insufficient activity | Sufficient activity | Insufficient activity |
| | n (%) | n (%) | n (%) | n (%) |
| **Gender** | | | | |
| Boy | 98 (40.7) | 143 (59.3) | 82 (34.0) | 159 (66.0) |
| Girl | 77 (32.9) | 157 (67.1) | 70 (29.9) | 164 (70.1) |
| **Weight status** | | | | |
| Underweight | 5 (35.7) | 9 (64.3) | 5 (35.7) | 9 (64.3) |
| Normal weight | 141 (44.9) | 173 (55.1) | 129 (41.1) | 185 (58.9) |
| Overweight | 25 (21.9) | 89 (78.1) | 17 (14.9) | 97 (85.1) |
| Obesity | 4(12.1) | 29 (87.9) | 1 (3.0) | 32(97.0) |

n: frequency; %: percentage; PAQ-C: Physical Activity Questionnaire for Older Children.

**Table 5. Measure of agreement between PAQ-C and pedometer on physical activity level (N = 475).**

| Variables | Pedometer | | Cohen's Kappa | p-value |
| --- | --- | --- | --- | --- |
| | Sufficient activity | Insufficient activity | | |
| | n (%) | n (%) | | |
| **PAQ-C** | | | 0.688 | **<0.001** |
| Sufficient activity | 130 (74.3) | 45 (25.7) | | |
| Insufficient activity | 22 (7.3) | 278 (92.7) | | |

n: frequency; %: percentage; PAQ-C: Physical Activity Questionnaire for Older Children.

time, leading them to focus on studying and being less physically active than usual. The result aligned with the research findings of Muhammad in Pakistan ($r_s = 0.883$) [60] and Wyszyńska's study on Polish adolescents ($r_s = 0.940$) [53]. However, our correlation was higher than that reported by Kathleen et al. ($r_s = 0.630$) [55] and Wang et al. ($r_s = 0.330$) [35] when assessing the correlation between triaxial accelerometer measurements and the modified version of the PAQ-C. Conversely, a systematic review conducted in 2008 showed low-to-moderate correlation coefficients observed between self-reported and objectively measured PA, ranging from 0.71 to 0.96. They also reported that no clear pattern emerged for the mean differences between self-report and direct measures of PA [61]. However, most reviews were conducted on adult populations, and there was a lack of reports carried out on young children.

PA is a multi-dimensional human behaviour that requires a multi-data approach [62]. Many methods can be used to measure PA, but no single method is considered the "gold standard" measure because each has its advantages and limitations. Our results showed that self-reported questionnaires may be a feasible alternative to other methods that are costly and involve complex monitoring processes. Nevertheless, it is important to combine both subjective and objective methods in research to provide a more comprehensive assessment of PA [16,63].

To the best of our knowledge, this study is the first to report validation of the Vietnamese version of the PAQ-C and examine the technical validity of both the PAQ-C and pedometer, providing deep insight into their accuracy in assessing PA in Vietnamese adolescents. Our findings provide a useful reference for further research in the future.

Several limitations should be considered in this study. First, the outcomes are limited to 12-year-old students from one high school, which restricts the generalizability of the findings to other age groups or broader adolescent populations. However, given the distinct developmental characteristics of this age group, the findings are particularly relevant to early adolescence. On the other hand, validating PA assessment instruments necessitates a controlled environment to minimize

external influences from environmental and curriculum-related factors, ensuring consistency in data collection. Previous validation studies have utilized similar school-based settings to enhance measurement reliability [39,64]. Second, the limitation of recall when using subjective self-report assessment tools is a challenge for all subjective measures. Nevertheless, this study used data collected from pedometers for comparison that can minimize recall bias. Third, pedometers have their limitations, as they cannot record activities in water and may be forgotten to be worn, especially by children. Therefore, it was difficult to ensure that pedometers were worn all through the study, which could have affected the accuracy of the PA measurements. Fourth, while the study assessed test-retest reliability with a sample of 35 students, aligning with sample size recommendations in established guidelines [26,65] and similar previous studies [35,40,66], we acknowledge that a larger sample could provide additional insights and strengthen the validity of results. Furthermore, the absence of a comprehensive content validation process may have introduced potential biases, thereby limiting the generalizability and practical applicability of the PAQ-C. Future research should employ standardized procedures to formally evaluate the content validity of the instrument, including the quantitative assessment of the relevance and clarity of each questionnaire item. Lastly, while Ho Chi Minh City has a stable climate with minimal extreme weather, our findings may not generalize to countries with distinct seasons or greater climate variability. In regions with harsh winters or extreme temperature shifts, such as North America, Europe, and East Asia, student PA levels may fluctuate considerably. Despite these limitations, our findings demonstrate that the PAQ-C is a highly reliable and stable instrument, suitable for broader population-based research.

## Conclusions

The study provided strong evidence of the reliability, internal consistency, and concurrent validity of the PAQ-C in 12-year-old Vietnamese adolescents, with substantial agreement between PAQ-C and pedometer data. Our findings suggest that PAQ-C is a practical and cost-effective tool for school-based PA monitoring, particularly in resource-limited settings. While PAQ-C may serve as a useful alternative, combining subjective and objective methods offers a more comprehensive assessment of PA. Future large-scale studies should explore the longitudinal use of PAQ-C alongside objective measures across diverse age groups to inform targeted interventions promoting PA in adolescents.

## Supporting information

**S1 Checklist. GRRAS checklist for reporting of studies of reliability and agreement.**
(DOCX)

**S1 Data. Data of the first sample (N = 35) and second sample (N = 475).**
(XLSX)

## Acknowledgments

The authors express their gratitude to the Department of Education and Training and the University of Medicine and Pharmacy at Ho Chi Minh City, Vietnam. We thank the principal and teachers at Nguyen Du Secondary School for their essential support in collecting samples, as well as the parents and students who participated in the study. We also acknowledge Master Warren Maurice Cowley from Saigon International University, Vietnam, and Phi Hoang Nguyen from University of Pennsylvania School of Dental Medicine, USA, for translating the questionnaire.

## Author contributions

**Conceptualization:** Tram T. N. Truong, Kien G. To.

**Data curation:** Tram T. N. Truong, Van-Anh N. Huynh.

**Formal analysis:** Tram T. N. Truong, Van-Anh N. Huynh, Kien G. To.

**Investigation:** Tram T. N. Truong.

**Methodology:** Tram T. N. Truong, Kien G. To.

**Project administration:** Tram T. N. Truong, Van-Anh N. Huynh, Kien G. To.

**Resources:** Tram T. N. Truong.

**Supervision:** Tram T. N. Truong, Kien G. To.

**Validation:** Tram T. N. Truong, Van-Anh N. Huynh, Kien G. To.

**Writing – original draft:** Tram T. N. Truong.

**Writing – review & editing:** Tram T. N. Truong, Van-Anh N. Huynh, Kien G. To.

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
