## [Decision Letter · Decision Letter 0]

19 Mar 2025

PGPH-D-25-00321

Comparison between self-reported and pedometer-measured physical activity in Vietnamese adolescents: A reliability and agreement study

Dear Dr. To,

Thank you for submitting your manuscript to PLOS Global Public Health. After careful consideration, we feel that it has merit but does not fully meet PLOS Global Public Health’s publication criteria as it currently stands. Therefore, we invite you to submit a revised version of the manuscript that addresses the points raised during the review process.

Please review the comments provided below and in the attached document and make the appropriate revisions to address their concerns.

We look forward to receiving your revised manuscript.

Kind regards,

Emma Campbell, Ph.D

Staff Editor

Journal Requirements:

Additional Editor Comments (if provided):

Reviewers' comments:

Reviewer's Responses to Questions

**Comments to the Author**

1. Does this manuscript meet PLOS Global Public Health’s publication criteria?

Reviewer #1: Yes

Reviewer #2: Yes

2. Has the statistical analysis been performed appropriately and rigorously?

Reviewer #1: Yes

Reviewer #2: Yes

3. Have the authors made all data underlying the findings in their manuscript fully available (please refer to the Data Availability Statement at the start of the manuscript PDF file)?

Reviewer #1: Yes

Reviewer #2: Yes

4. Is the manuscript presented in an intelligible fashion and written in standard English?

Reviewer #1: Yes

Reviewer #2: Yes

Reviewer #1: Thank you for allowing me to review this paper. This paper is well written and well organised. There is always value in considering fit and cultural adaptations in questionnaire measurement instruments generally and I commend you on the care and attention to detail that you have undertaken in your work.

I have a number of questions that I would like you to consider and address in your manuscript.

1. Explain which ‘expert groups’ are you referring to in the Introduction (page 4, line 62), this should include references to support.

2. Is there an effect of the 3 month time period March-May that you chose for data collection both in terms of seasonality of weather (e.g., inclement weather affecting physical activity; PA) and also in terms of the school year (e.g., academic activities)?

3. Why one school only? Please describe this school more thoroughly for an international audience. For example, is it placed in an area of high/low socio-economic deprivation/affluence, is it a public or a private school? These factors can contribute to physical activity generally and especially in terms of children.

4. The PA programme that is described in the school – is this a normal programme or enhanced as a result of the research project taking place in the school?

5. Did you do a power calculation to determine the necessary sample size for the project?

6. Greater justification is needed for not using an accelerometer (which would consider intensity, duration and type of PA) and for pedometer use.

7. Did the children have access to their step count at the end of each day? A procedure point notes that the children reset the pedometer themselves. This would suggest that they could also see their daily steps. If this is the case, please comment on how you believe this could have enhanced their ability to then more accurately report their step count in the self-reported survey? Were any attempts made to ‘blind’ the children to their objectively measured daily step count?

Reviewer #2: Comments

1. Questionnaire: What is the content validity index of the questionnaire?

2. What is the brand of the pedometer? What is its validity and reliability?

3. How did you ensure that students did not manipulate the pedometer results, such as shaking it?

4. Why were results less than 1000 and more than 30,000 per day excluded? Please cite a reference for this.

5. Please add sample size calculations for Phase 1 and Phase 2.

6. Define sufficient and insufficient physical activity in the PAQ-C and pedometer sections

Statistical Analysis

1. Lines 185 -186 should be part of the method.

2. Line 199: Cohen’s Kappa test was calculated to measure the level of agreement between the two methods

Can you give classification of interpreting results.

Ethical considerations

1. This should be part of the method section.

2. Was the assent form obtained from children?

Results

1. Please include a flow diagram.

2. Kindly include gender results in the first paragraph of the result section

**Do you want your identity to be public for this peer review?** For information about this choice, including consent withdrawal, please see our Privacy Policy

Reviewer #1: No

Reviewer #2: No

---

## [Decision Letter · Decision Letter 1]

14 May 2025

Comparison between self-reported and pedometer-measured physical activity in Vietnamese adolescents: A reliability and agreement study

PGPH-D-25-00321R1

Dear Dr. To,

We are pleased to inform you that your manuscript 'Comparison between self-reported and pedometer-measured physical activity in Vietnamese adolescents: A reliability and agreement study' has been provisionally accepted for publication in PLOS Global Public Health.

Best regards,

Julia Robinson

Executive Editor

Reviewer Comments (if any, and for reference):

Reviewer's Responses to Questions

**Comments to the Author**

Reviewer #1: All comments have been addressed

publication criteria?

Reviewer #1: Yes

3. Has the statistical analysis been performed appropriately and rigorously?

Reviewer #1: Yes

4. Have the authors made all data underlying the findings in their manuscript fully available (please refer to the Data Availability Statement at the start of the manuscript PDF file)?

Reviewer #1: Yes

5. Is the manuscript presented in an intelligible fashion and written in standard English?

Reviewer #1: Yes

Reviewer #1: I am satisfied that the authors have addressed all comments that I raised.

**Do you want your identity to be public for this peer review?** For information about this choice, including consent withdrawal, please see our Privacy Policy

Reviewer #1: No
